REGISTERED REPORT PROTOCOL

# Are quality assessments in science affected by anchoring effects? The proposal of a follow-up study

**Lutz Bornmann**[1]*, **Christian Ganser**[2]

**1** Science Policy and Strategy Department, Administrative Headquarters of the Max Planck Society, Munich, Germany, **2** Department of Sociology, Ludwig-Maximilians-Universität Munich, Munich, Germany

* bornmann@gv.mpg.de

This is a Registered Report and may have an associated publication; please check the article page on the journal site for any related articles.

**Data Availability Statement:** Only the results of simulations are reported in the manuscript.

## Abstract

We plan to empirically study the assessment of scientific papers within the framework of the anchoring-and-adjustment heuristic. This study is a follow-up study which is intended to answer open questions from the previous study with the same topic Bornmann (2021) and Bornmann (2023). The previous and follow-up studies address a central question in research evaluation: does bibliometrics create the social order in science it is designed to measure or does bibliometrics reflect the given social order (which is dependent on the intrinsic quality of research)? If bibliometrics creates the social order, it can be interpreted as an anchoring-and-adjustment heuristic. In the planned study, we shall undertake a survey of corresponding authors with an available email address in the Web of Science database. The authors are asked to assess the quality of articles that they cited in previous papers. The authors are randomly assigned to different experimental settings in which they receive (or not) citation information or a numerical access code to enter the survey. The control group will not receive any further numerical information. In the statistical analyses, we estimate how (strongly) the quality assessments of the cited papers are adjusted by the respondents to the anchor value (citation counts or access code). Thus, we are interested in whether possible adjustments in the assessments can not only be produced by quality-related information (citation counts), but also by numbers that are not related to quality, i.e. the access code. Strong effects of the anchors would mean that bibliometrics (or any other number) create the social order they are supposed to measure.

## 1 Introduction

Tversky and Kahneman [1] call a phenomenon as anchoring and insufficient adjustment when initial values are used by humans as starting points in estimations: different numerical starting points lead to different numerical estimates. In order to demonstrate the effect of anchoring, Tversky and Kahneman [1] asked interviewed persons to estimate the percentage of African nations in the United Nations (UN). In the persons' presence, a wheel of fortune was spun to receive a random number between 0 and 100. In the first instruction, the

**Funding:** The authors received no specific funding for this work.

**Competing interests:** The authors have declared that no competing interests exist.

interviewed persons should indicate whether the random number is higher or lower than the African nations' percentage "by moving upward or downward from the given number" [1]. In the second instruction, they were requested to estimate the percentage. The results of the experiment revealed a significant effect of the random numbers on the estimates: "For example, the median estimates of the percentage of African countries in the United Nations were 25 and 45 for groups that received 10 and 65, respectively, as starting points" [1].

Since the study of Tversky and Kahneman [1], many studies have investigated anchoring effects in various situations (of estimation, assessment or decision). Overviews of these studies can be found in Mussweiler, Englich [2], Furnham and Boo [3], Bahnik, Englich [4], and Bahník, Mussweiler [5]. The literature overviews come to similar conclusions with respect to the prevalence of anchoring effects in various contexts. According to Mussweiler, Englich [2], for example, "anchoring effects are among the most robust and easily replicated findings in psychology" (p. 184). Furnham and Boo [3] concluded as follows: "research in the field demonstrates that anchoring is a pervasive and robust effect in human decisions" (p. 41). The authors of a meta-analysis [6] including studies that have investigated anchoring effects write: "anchoring has a significant impact on negotiators and negotiation outcomes" (p. 598). Previous research even demonstrated that anchoring effects appeared in situations in which persons were informed about the anchors and instructions were used to correct anchoring effects.

Since assessments (under uncertainty) are prevalent in research evaluations, Bornmann, Ganser [7] studied for the first time whether anchoring effects also exist in the area of research assessments. The studying of anchoring effects is important, since quality assessments should be based on research itself, and not on (irrelevant) numbers which may bias the assessments. Following Traag and Waltman [8], we define bias as a direct causal effect of one variable on another variable that is unjustified (e.g., the effect of gender on scientific assessments). If scientific assessments are dependent on (irrelevant) anchors, the assessments are possibly biased and their validity can be questioned. The study of Bornmann, Ganser [7] applied a similar study design as Teplitskiy, Duede [9]–although Teplitskiy, Duede [9] did not explicitly investigate anchoring effects. Bornmann, Ganser [7] undertook a survey of corresponding authors from the Web of Science (WoS) database (Clarivate) whom they presented various information as possible anchors.

In the study of Bornmann, Ganser [7], the corresponding authors were requested to assess the quality of articles that they cited in previous papers. The authors were randomly assigned to three treatment groups and a control group. The treatment groups received the following information alongside the title and abstract of the cited paper: paper impact information, impact information on the publishing journal, and a numerical access code to enter the survey. The control group did not receive any information besides title and abstract. Bornmann, Ganser [7] explored on the one hand whether possible adjustments in the assessments of cited articles can be produced by information (paper impact or journal impact) that are related to quality. Their results revealed then that the quality assessments of papers seem to depend on the paper, but not the journal impact information. The significant results on the paper impact information confirm similar results by Teplitskiy, Duede [9]. Bornmann, Ganser [7] explored on the other hand the effect of numbers on quality assessments that are not related to quality, i.e. access codes to the survey. With respect to the access codes, the results indicated that these arbitrary numbers do not play a role in the assessments of papers.

In this registered report protocol, we present the proposal for a follow-up study that is intended to investigate open questions from the results of Bornmann, Ganser [7]. One open question refers to the paper impact information that revealed a statistically significant effect in the study by Bornmann, Ganser [7]. The effect may be interpreted as a confirmation of the paper impact information as anchor whereby quality adjustments were caused by anchors.

The results of Bornmann, Ganser [7] yet do not allow this causal conclusion, since the results are based on a limited design: the respondents in the groups (receiving paper impact information or not) assessed different cited articles. Not only the paper impact information was varied in the study, but also the assessed article (and, by association, the article's quality may have also varied). For causal conclusions, it is necessary that randomly selected respondents assess the same cited paper under two conditions: with (different) or without paper impact information.

Another open question of the study by Bornmann, Ganser [7] concerns the investigated influence of the access code on the quality assessments. The found missing effect of the access code in the study may result from the handling of the access code in the questionnaire. Since Bornmann, Ganser [7] only requested the respondents to fill in the access code in the questionnaire, respondents were not demanded to engage more deeply with the access code. It is planned therefore in the follow-up study to stimulate this engagement. Following the classical study by Tversky and Kahneman [1], it is planned that respondents assess in a first step whether the quality assessment (the quality code) is higher or lower than the access code. In the second step then, the absolute quality assessments follow–as was done by Bornmann, Ganser [7]. We expect that the engagement of the respondents with the access code will be intensified by asking for the relative assessment in the first step, and an effect will be possibly evoked on the absolute quality assessment in the second step. This stimulation of the engagement is also planned for the citation counts presented to the respondents (to have the same experimental conditions for all respondents).

Before we explain the planned research design in the following, we present a literature overview of the anchoring-and-adjustment framework. This overview is kept short since the relevant literature has been recently summarized in Bornmann, Ganser [7].

## 2 Literature overview

This literature overview is mainly based on the overviews by Mussweiler, Englich [2], Bahnik, Englich [4], and Bahník, Mussweiler [5]. Anchoring can be defined as "the assimilation of a numeric judgement to a previously considered standard"[2]. An anchoring effect has been detected in different judgement situations (e.g., knowledge questions, price estimates, and legal judgements). Anchoring effects have been demonstrated in absolute judgments (e.g., how great is something) and in comparative judgments (e.g., is something smaller or larger). Anchors may be of different type: they can be numeric values or specific text stimuli. Effects of anchors have been generated in laboratory settings [e.g., 1] and in 'real-world' settings [e.g., 10].

Anchoring effects have been investigated in the context of four experimental paradigms, in which "the anchor values are either explicitly or implicitly provided by the experimenter, self-generated, or provided in an unrelated task" [2]. The paradigm in many studies is the approach by Tversky and Kahneman [1] where the anchor values are provided in an unrelated task including first comparative and then absolute anchoring questions. The planned study is oriented towards this usual approach.

Studies in the four experimental paradigms have revealed that anchoring effects are robust with respect to various moderating variables. Anchors can be differentiated whether or not they are relevant for a judgmental task to be effective. In the study by Bornmann, Ganser [7], e.g., citation information can be interpreted as relevant for quality assessments of papers, but access codes as not relevant. One may assume that only relevant anchors are effective in assessments. Furnham and Boo [3] concluded yet in their overview of the literature that "irrelevant anchors produce similar effects in judgmental decisions in comparison to those of informational relevance anchors" (p. 38). Research also demonstrated that anchoring effects are

effective in situations in which humans tried to work against the influence of presented anchors (e.g., by using instructions in the study to correct for an influence of the anchor). Extremity and implausibility of anchors also do not seem to prevent anchoring of being effective.

Five theoretical backgrounds have been proposed for explaining anchoring effects: it has been suggested that "anchoring effects result from (1) insufficient adjustment from an anchor, (2) conversational inferences, (3) numerical priming, (4) mechanisms of selective accessibility, and (5) distortion of the response scale"[5].

## 3 Citation counts as possible anchors in quality assessments

Doing science is not imaginable without research evaluation: "scientific progress is nowadays strongly dependent on research evaluation processes, as they regulate the stream of ideas and research projects by means of science funding allocation" [11]. The use of citations for research evaluation has a long tradition: the term evaluative bibliometrics was introduced by Narin [12] decades ago based on the structural-functional approach by Merton [13]. Citation analyses are (currently) used for the evaluation of research groups, institutions, and countries as well as of research proposals and hiring of academic personnel [14]. For example, Sivertsen [15] published an overview of the use of various bibliometric indicators in national performance based research funding systems of several countries such as Italy and UK. Bibliometrics has this prominent role in research evaluation, since it combines four advantages [16]: (1) bibliometrics is able to provide single numbers that may reflect research performance; (2) bibliometrics may provide performance numbers that are comparable across disciplines (by the use of field-normalized indicators); (3) bibliometrics can be used unobtrusively with available process-oriented numbers from large databases; (4) the costs of using bibliometrics are (usually) low when compared with the use of peer review panels.

The frequent use of bibliometrics in research evaluation is accompanied by massive critique–despite its many advantages. An overview of important critical points can be found in Jappe, Pithan [17]. In the planned study, we will address one of these points: does bibliometrics create the social order in science it is designed to measure? If so, the social order may be independent from the intrinsic quality of research. The opposite assumption is that bibliometrics reflects the given social order (which is dependent on the intrinsic quality of research). If bibliometrics creates the social order, it can be interpreted as an anchoring-and-adjustment heuristic [1]. Bibliometrics would then mean a starting point of an evaluator's assessment about a certain piece of research.

## 4 Study design

In the planned follow-up study, we will investigate anchoring effects in the assessment of cited papers–as in Bornmann, Ganser [7]. The planned study is intended to focus stronger on possible causal relations between quality assessments and information presented to interviewed persons [8] and to investigate more rigorously possible anchoring effects. The description of the study design mainly follows the description of the previous study [7] by Bornmann, Ganser [18]. In the follow-up study, we will undertake a survey of corresponding authors with an available email address in the Web of Science database (a multi-disciplinary literature database including citations). Our questionnaire and the cover letter can be found in the S1 Appendix.

The corresponding author will receive an email with a link to a web-based questionnaire. In the email, the authors are informed about the topic of the study and the source of their email addresses. Furthermore, they are told that the collected data will only be analyzed in anonymized form. If the authors decide to take part in the survey, they are asked for the assessment

of a paper (reference) that they cited in a previous publication. The authors are randomly assigned to five different experimental settings in which they receive (or not) (1) information about the (true) citation impact of the paper (as an anchor with possibly relevant information for the quality assessment; the author is informed about the use of the citation impact as possible anchor in this study), (2) information about the (true) citation impact of the paper (the author is not informed about the use of the citation impact as anchor), (3) a randomly generated anchor (an access code to the questionnaire) that is not related to the quality of publications (the author is informed about the use of the access code as anchor) or (4) a randomly generated anchor (an access code to the questionnaire; the author is not informed about the use of the access code as anchor). A fifth group of authors does not receive any additional information that may function as anchor.

When the respondents decide to take part and start the survey, they will be asked whether they remember the presented paper (or not). Further questions refer to how well the authors know the paper ('extremely well' through to 'not well'), how much the cited paper influenced the research choices in the citing paper, and which aspects of the citing paper were influenced by the cited paper. In the final part of the questionnaire, the respondents rate the cited paper against possible others in the field concerning several characteristics (quality, novelty, significance, validity, generalizability, and canonical reference). The range of the rating scale will be from 1 (bad) to 100 (excellent). The respondents who are not informed at the beginning about the use of access code or citation counts as potential anchors are informed about the design of the study at the end of the questionnaire. This means that all respondents will be informed about the aim of the study (either at the beginning or end of the questionnaire).

In the planned study, we will mainly replicate the survey design by Bornmann, Ganser (1). The following changes are planned yet:

1. <u>Assessments of highly-cited papers</u>: We will include in the follow-up study cited papers from 2010 that belong to the 1% most frequently cited papers in their publication year and subject category (only papers with the document type "article"). Since highly-cited papers receive many citations and we plan to survey citing authors, we assume that we will receive sufficient filled-in questionnaires with assessments of one and the same paper. Although we also planned in Bornmann, Ganser [18] to have multiple assessments for single papers, too few filled-in questionnaires resulted from the survey to consider multiple assessments in the statistical analyses. This may change with the focus on highly-cited papers with their many citing papers. Another advantage of highly-cited papers besides the availability of many citing papers is that highly-cited papers may be well-known to the citing authors. A good knowledge of the cited paper is an important condition for an informed quality assessment by the citing authors. The focus on highly-cited papers leads, however, to a limitation in the representativeness of the planned study's results. To ensure that we receive enough responses (assessments of one and the same paper), we abstain from including lowly-cited papers. The results of the planned study will be representative, thus, only for the assessments of highly-cited papers.

   In the planed study, each cited paper will be assessed under four treatment group conditions and the control group condition. This enables us to control the effect of the cited paper in the statistical analyses (by using models for repeated measures).

2. <u>Consideration of subject categories</u>: Citations have different meanings in different disciplines. In one discipline, citation counts are frequently used for research evaluation purposes; in other disciplines, the use of these counts are uncommon [19]. In order to consider these disciplinary differences in the interpretation of the results from the planned study, the subject categories of the cited and citing papers will be considered in the questionnaire.

Filled-in questionnaires will include then the subject categories that can be considered in the statistical analyses to investigate whether effects differ between the categories.

3. Paper impact information: In the previous study of Bornmann, Ganser [7], the paper impact information was presented to respondents as percentile impact value [20]. Since the respondents assessed cited papers from different disciplines, it was necessary to standardize the citation impact information using percentiles. Although the planned follow-up study also includes papers from different disciplines, it is the objective that each cited paper is assessed multiple times (by different respondents). This study design leads to the comparison of assessments within single cited papers, but not between different cited papers. Therefore, it is not necessary to normalize the citations based on percentiles. The respondents who will receive the paper impact information will be presented one of the following citation counts: (i) number of citations from the first three years after publication or (ii) number of citations from the first five years after publication. These citation information are available in the in-house database that we used as bibliometric database for the planned study. By using different citation counts for the same cited paper, we can study the influence of different quality-related anchors on the quality assessment of the same paper: do the assessments vary depending on the presented anchor, which is randomly assigned to respondents?

4. Randomly generated anchor with irrelevant information: In the planned study, we shall consider an anchor with irrelevant numerical information [see 21]. This anchor will be presented as an access code in the cover letter. The respondents are asked for copying the code in the web-based questionnaire [see here 22]. The access code will be randomly drawn from a uniform distribution.

5. Absolute and relative quality assessments: In Bornmann, Ganser [7], the effect of the access code on the quality assessments was statistically not significant. The missing effect may result from a limited study design that only focusses on an absolute quality assessment. In the classical study by Tversky and Kahneman [1], interviewed persons first judged relatively against the anchor, before they made the absolute assessment. Following this classical study design, we will ask the corresponding authors in our survey first whether their quality assessment is higher or lower than the access code and second to fill-in their quality assessment score in the questionnaire. In order to have the same experimental conditions for all respondents in our survey (in the treatment groups), the relative and absolute assessments are also demanded from respondents who are confronted with citation counts.

6. Range of citation counts and access codes: The range of the quality assessment scores in the planned study is from 1 to 100. The citation counts and access codes that are used in this study will be in the range of 2 to 99. The slight range differences between the scales are necessary to allow lower or higher quality assessments than citation counts and access codes. The possibility of higher or lower assessment scores is demanded for the relative assessments of the respondents (see above). Many highly-cited-papers in our sample have higher citation counts than 99, i.e. higher than the range of possible quality assessment scores minus one. Some highly-cited papers have no or only one citation in the first three (five) years after publication, i.e. 33 (9) highly-cited papers. In order to prevent the possibility that the citation counts are out of the range (with respect to the corresponding quality assessment scores), we decided not to consider cited papers in the planned study with citation counts lower than 2 and higher than 99.

7. <u>Unmasking the anchor</u>: Research on anchoring effects has shown that anchors are effective although the interviewed persons were informed about the use of anchors in the study [2]. Therefore, we plan to inform a part of our interviewed persons about the study design and the use of access code and citation counts as anchors.

8. <u>Proximity and distance of respondents to research reported in the cited paper</u>: We plan to survey authors who cited certain papers in the past. We assume that authors who are active in similar research areas as the research reported in the cited papers are in a better position to assess the cited paper than authors who are active in research areas that are more distant. In order to control proximity and distance in the statistical analyses, we will request the respondents to assess the proximity and distance of their own research to the research reported in the cited paper–on a scale from 1 (very close) to 100 (far away).

9. <u>Search for citation counts by respondents themselves</u>: We cannot preclude that respondents themselves (in the experimental groups and control group) search for citations of the presented highly-cited paper. In order to consider the possibility of these searches in the study design and to control potential distortions in the statistical analyses, we will ask the respondents whether they searched for citation counts (and if so, which number they found).

The bibliometric data that we shall use in our study originate from two databases: (1) Web of Science data from an in-house database developed and maintained by the Competence Center for Bibliometrics (CCB, see also https://bibliometrie.info/, supported by BMBF grant 16WIK2101A); (2) an in-house database developed and maintained by the Max Planck Digital Library (MPDL, Munich) and derived from the Science Citation Index Expanded (SCI-E), Social Sciences Citation Index (SSCI), Arts and Humanities Citation Index (AHCI) prepared by Clarivate. We focus on cited papers published in 2010, since we are interested in corresponding authors from many citing papers of the same cited paper. In September 20, 2023, we downloaded 8,620 highly-cited papers published in 2010 (papers that belong to the 1% most frequently cited in the corresponding subject category and publication year). Of these papers, we will consider only articles (on the cited and citing side), since different types of documents may lead to different selection decisions of cited papers by citing authors at that time. For reviews, cited papers were selected (and read) to be included in an overview of research; for articles, cited papers were more likely to be the primary starting point for own research.

In our study, articles are only considered if they have an email address for the corresponding author in the database (some papers in the database do not have an email address). The authors of the citing papers may also be the authors of the cited papers in the planned survey. In order to receive the information whether an author assesses an own previous paper or not in the survey, we will ask the respondents whether the presented paper in the questionnaire were published by themselves. Since we assume that own previous papers are well-known, the assessment of these papers will be based on a well-informed basis. Although different citing papers may have the same email address, each email address will be selected not more than once.

The selection process of cited and citing papers resulted in 1,394,099 email addresses and 2,545,239 citing papers. Each of the corresponding authors will be assigned randomly to one of the four treatment groups or the control group. Each treatment group receives one of both items of information: citation counts (for the cited article) or access code. The treatment groups are informed or not informed about the use of citation counts or access codes as anchors in the survey (at the beginning). The authors in the control group will not receive any of this information. This will result in 278,819 authors for each group. Bornmann, Ganser [7] sent an email to 77,872 addresses. About 2% reached the last page of their survey. Our power analysis (see section 6) shows that we need a maximum of 3,600 respondents. If we have a

similar response rate in the follow-up study, a total of 180,000 valid email addresses would be sufficient to reach this target. In the planned study, the email addresses will only be used for inviting the respondents. Many addresses contain the name of the owner and possibly allow the identification of participants. After collecting the data, however, the email addresses will be deleted and the anonymized data analyzed. The anonymized data will not allow the identification of respondents. Access to the anonymized data will be provided via figshare with the DOI 10.6084/m9.figshare.24182238 (which we already reserved).

In our sample, we will have corresponding authors with different levels of experience in their field. We anticipate that the assessments of cited papers are dependent on this level: senior researchers may be in a better position to assess the quality of a cited paper than junior researchers. There may be the risk, therefore, that junior researchers are more open to biases in their assessments. In the planned study, we target these differences between the corresponding authors interviewed by means of their random assignments to the treatment and control groups. Furthermore, the literature overview by Furnham and Boo [3] shows that "expertise does not significantly reduce the assimilative bias in decisions that affect inexperienced laypeople" (p. 39). Similar results have been published by Bystranowski, Janik [23].

## 5 Statistical analysis

The web-based survey will yield quality assessments of cited articles (concerning quality, novelty, significance, validity, generalizability, and canonical reference) on a percentile scale from 1 (very poor) to 100 (exceptional) whereby 50 denotes a typical paper. These quality assessments are the dependent variable, and we are interested in how they relate to the independent variables: citation information and access code, either revealed to the respondents as an anchor or not. Our dataset consists of repeated measures, since each cited paper will be assessed under up to five conditions (four treatments): by presenting citation information (revealed/not revealed to the respondents at the beginning as an anchor), access code (revealed/not revealed to the respondents as an anchor) or no further information. We will demonstrate the planned statistical analyses in the following using fictitious data.

Table 1 shows fictitious data (from a web-based survey) for a cited article (article 1). The cited article receives 87 citations. This information is presented to respondent 1. Respondent 2 enters the survey with the access code 89 without knowing that the code is used as an anchor. Respondent 3 knows that the access code 92 (she entered) serves as an anchor. Respondent 4 assesses the cited reference without indicating an access code or receiving citation impact information.

We plan to analyze the data with multilevel models for repeated measures (i.e. at different conditions). We shall estimate one model for each quality assessment of cited articles. In each model,

**Table 1. Fictitious data that are paper-based (e.g., citation counts) or respondent-based (e.g., quality assessments) for cited article 1.**

|  | Citations (not revealed) | Citations (revealed) | Access code (not revealed) | Access code (revealed) | No information |
|---|---|---|---|---|---|
| **Paper data** |  |  |  |  |  |
| Value | 87 | 87 | 89 | 92 |  |
| **Survey data from five respondents** | 1 | 2 | 3 | 4 | 5 |
| Quality | 35 | 30 | 25 | 45 | 20 |
| Novelty | 30 | 33 | 45 | 22 | 35 |
| Significance | 21 | 25 | 32 | 78 | 74 |
| Validity | 34 | 40 | 47 | 57 | 76 |
| Generalizability | 44 | 50 | 55 | 83 | 42 |
| Canonical reference | 20 | 18 | 12 | 15 | 95 |

we examine how the various information at a certain condition (e.g., different citation counts at the citation impact condition) is related to the quality assessments. Fictitious data for three cited articles (including cited article 1 from Table 1) are reported in the long format in Table 2.

We examine treatment effect heterogeneity by estimating fixed effects regression models, including the values of the treatments. As shown by Bornmann, Ganser [7], citations are positively correlated with quality assessments even if they were not presented to the respondents. Therefore, we include citation counts for the respondents in the control group. This allows us to compare effect sizes of the treatments with the control condition. The model is specified as follows:

$$quality\ assessment_{ij}$$
$$= \beta_0 + \beta_1 condition2_{ij} + \beta_2 condition3_{ij} + \beta_3 condition4_{ij}$$
$$+\beta_4 condition1 \cdot value + \beta_5 condition2 \cdot value + \beta_6 condition3 \cdot value$$
$$+\beta_6 condition4 \cdot value + u_j + e_{ij},$$

where *conditions 1* to *4* are the three treatment conditions and the control condition and *value* is the value of the respective information presented to the respondents.

In order to facilitate the interpretation of the results, this model does not include a main effect for the value, but four interactions. The coefficients of the interaction terms represent the effects of the respective value on the quality assessment for the four treatments. For example, $\beta_4 = .095$ means that an increase in the citation counts by 10 increases the quality assessment by .95. Exemplary predicted values from the model (which are not affected by the inclusion or non-inclusion of the main effect) are shown in Figs 1– 3 in the next section, where we present results of power analyses.

Data from the further questions in the web-based survey (i.e. whether the authors know the paper, how much the cited reference influenced the research choices in the citing paper, and

**Table 2. Fictitious data in the long format for one quality dimension.**

| Cited article | Respondent | Condition | Value | Quality assessment |
|---|---|---|---|---|
| 1 | 1 | 1 | 87 | 35 |
| 1 | 2 | 2 | 87 | 25 |
| 1 | 3 | 3 | 89 | 45 |
| 1 | 4 | 4 | 92 | 20 |
| 1 | 5 | 5 | | 37 |
| 2 | 6 | 1 | 61 | 60 |
| 2 | 7 | 2 | 93 | 41 |
| 2 | 8 | 3 | 22 | 20 |
| 2 | 9 | 4 | 45 | 20 |
| 2 | 10 | 5 | | 35 |
| 3 | 11 | 1 | 75 | 50 |
| 3 | 12 | 2 | 80 | 61 |
| 3 | 13 | 3 | 35 | 36 |
| 3 | 14 | 4 | 37 | 65 |
| 3 | 15 | 5 | | 60 |

Note. 1 = Citation information (not revealed), 2 = Citation information (revealed), 3 = Access Code (not revealed), 4 = Access code (revealed), 5 = No information

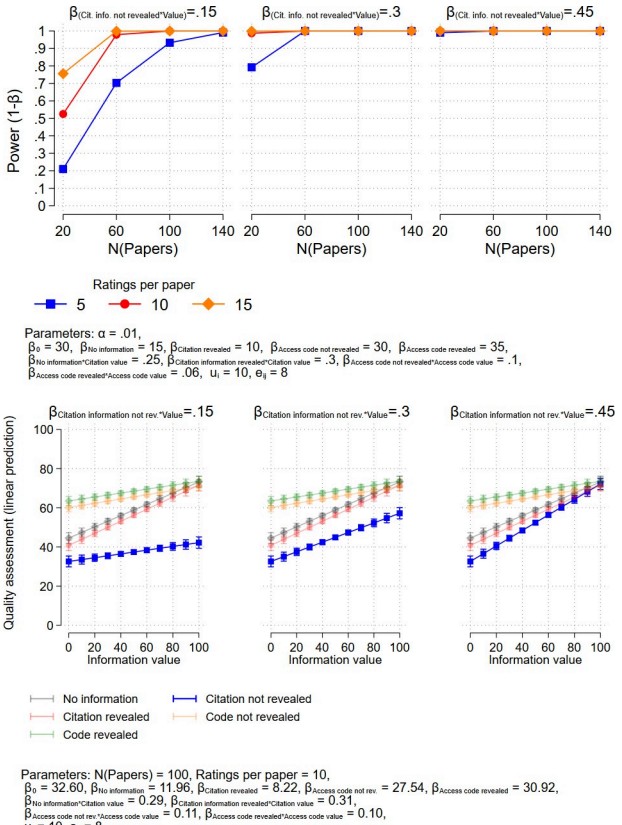

**Fig 1.** Upper panel: Power analysis (two-sided test) for the effect of presenting citation information. Lower panel: Predicted values from corresponding exemplary regression models.

which aspects of the citing paper were influenced by the cited reference) will be entered as possible control variables in the regression models.

## 6 Power analysis

How many respondents will be necessary to reject the null hypotheses when they are false? To answer this question, we present results from a power analysis [24] based on the results presented in Bornmann, Ganser [7]. The results of our previous study provide useful hints on the effect sizes we can expect in the planned study. We followed the simulation procedure outlined in Huber [25], Huber [26], Huber [27], Huber [28], and Raoniar [29] and made the following assumptions:

1. The variable including three-year-citation counts in our dataset, truncated to a range from 2 to 99, approximately follows a gamma distribution. Therefore, we simulated citation counts following a gamma distribution, rounded to integers, and truncated to the range from 2 to 99 [30]. The variables were generated in Python using the SciPy sub-package scypi.stats with parameters a = 2.62, loc = -0.8, and scale = 15.13. Truncated five-year-citation counts were simulated by using a beta distribution with parameters a = 1.96, b = 1.25, loc = 0.11, and scale = 99.18. For the regression models underlying the power analysis, one of the two resulting citation counts (referring to the three or five year citation window) was chosen randomly.

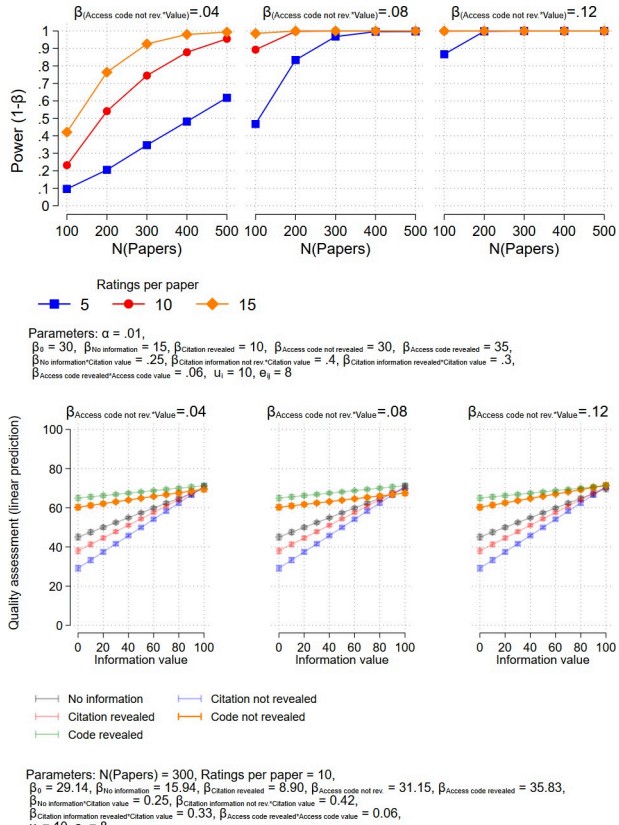

**Fig 2.** Upper panel: Power analysis (two-sided test) for the effect of presenting an access code without revealing it's use as an anchor to the respondents. Lower panel: Predicted values from corresponding exemplary regression models.

2. Regression parameters were chosen in a way resembling the results presented in Bornmann, Ganser [7]. In particular, we assumed stronger effects for citation information as for access codes. Furthermore, we suspected that citation counts and access codes have a stronger effect on quality assessments when respondents are not aware of their use as anchor.

3. We set the number of respondents rating each paper to multiples of 5 –the number of groups (experimental and control) in our study. We assumed that the respondents are evenly distributed over the five groups and the highly-cited papers under consideration.

4. The standard deviation of error terms $e_{ij}$ was set to 8; the standard deviation of error terms $u_j$ was set to 10.

5. The level of significance $\alpha$ was set to 0.01. We conducted separate analyses for the respective independent variables. Since our models include four interaction effects, we take into account the growing probability of getting statistically significant results by chance. Setting $\alpha$ to 0.01 ensures sufficient numbers of cases in a joint analysis for $\alpha = 0.05$. Our desired power $1 - \beta$ is 0.8.

We conducted analyses for investigating the effects of citation information and access codes on the overall quality assessments. We expected that the effect of citation information where the anchor is revealed to the respondents (effect A) lays between (1) the effect of citation information where the anchor is not revealed (effect B) and (2) the effect of the access code (effect C). Since the number of cases needed to detect effect A is between the cases needed to detect

effects B and C, we present results only for the following groups: (1) the citation information where the anchor is not revealed, (2) the access code where the anchor is not revealed, and (3) the access code where the anchor is revealed. The number of cases needed to detect effect B is expected to be between the cases needed to detect the effects for the three groups. The results of the power analyses are as follows:

To be able to detect effects of citation information where the anchor is not revealed to the respondents, we only need about 20 (papers) • 5 (respondents per paper) = 100 to 80 • 4 = 320 respondents, depending on the true effect size (see Fig 1).

To detect an effect of the access code where the anchor is not revealed to the respondents, more cases are needed since we expect effect sizes to be weaker. About 210 • 15 = 3150 respondents would be necessary to find the weakest effect assumed in our analysis, while about 130 • 5 = 650 respondents would be sufficient to find the strongest effect that we assume (see Fig 2).

We suppose that it will be even more difficult to detect the effect of the access code which is revealed to the respondents as an anchor, since we expect the smallest effect size for this treatment. Since we have many citing papers in our dataset, it will be possible to achieve the necessary sample size of about 300 • 15 = 4.500 respondents. This number would be needed at least to find the weakest effect studied (see Fig 3). For the strongest effect under consideration, about 1.000 respondents would be sufficient.

As the results of the power analyses show, collecting many ratings per highly-cited paper is crucial for our planned study. Since we expect that not every paper will receive the same

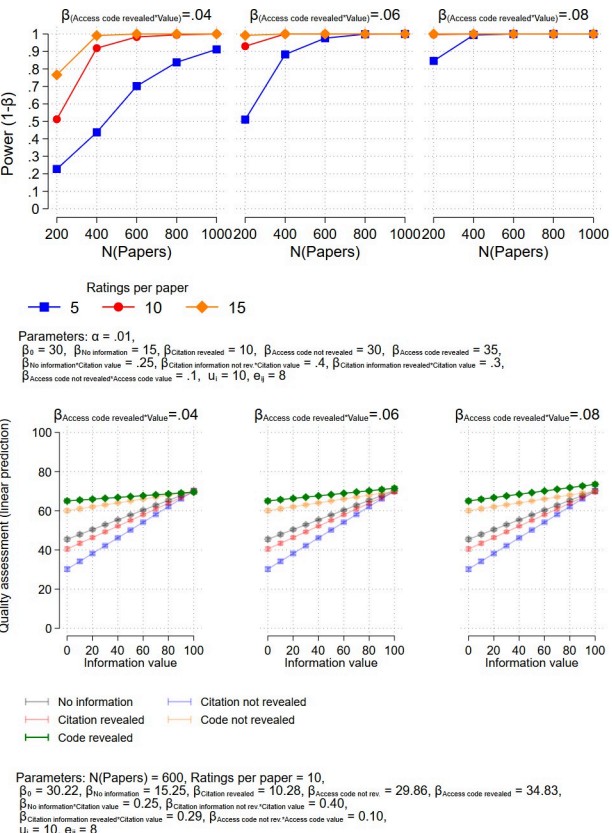

**Fig 3.** Upper panel: Power analysis (two-sided test) for the effect of presenting an access code and revealing it's use as an anchor to the respondents. Lower panel: Predicted values from corresponding exemplary regression models.

number of ratings, the results of the power analyses confirm that our approach of using highly-cited papers is reasonable.

## Supporting information

**S1 Appendix.**
(DOCX)

## Author Contributions

**Conceptualization:** Lutz Bornmann, Christian Ganser.

**Formal analysis:** Christian Ganser.

**Investigation:** Lutz Bornmann, Christian Ganser.

**Methodology:** Lutz Bornmann, Christian Ganser.

**Validation:** Christian Ganser.

**Writing – original draft:** Lutz Bornmann, Christian Ganser.

**Writing – review & editing:** Lutz Bornmann, Christian Ganser.

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
