## [Decision Letter · Decision Letter 0]

13 Sep 2023

PONE-D-23-23451Are quality assessments in science affected by anchoring effects? The proposal of a follow-up studyPLOS ONE

Dear Dr. Bornmann,

Thank you for submitting your manuscript to PLOS ONE. After careful consideration, we feel that it has merit but does not fully meet PLOS ONE’s publication criteria as it currently stands. Therefore, we invite you to submit a revised version of the manuscript that addresses the points raised during the review process.

All reviews have been positive. However, I kindly suggest addressing the recommendations provided by Reviewer 1. Consequently, I recommend submitting a minor revision.

We look forward to receiving your revised manuscript.

Kind regards,

Filipi Nascimento Silva

Academic Editor

PLOS ONE

Journal Requirements:

3. In your cover letter, please confirm that the research you have described in your manuscript, including participant recruitment, data collection, modification, or processing, has not started and will not start until after your paper has been accepted to the journal (assuming data need to be collected or participants recruited specifically for your study). In order to proceed with your submission, you must provide confirmation.

Reviewers' comments:

Reviewer's Responses to Questions

**Comments to the Author**

1. Does the manuscript provide a valid rationale for the proposed study, with clearly identified and justified research questions?

Reviewer #1: Yes

Reviewer #2: Yes

2. Is the protocol technically sound and planned in a manner that will lead to a meaningful outcome and allow testing the stated hypotheses?

Reviewer #1: Yes

Reviewer #2: Yes

3. Is the methodology feasible and described in sufficient detail to allow the work to be replicable?

Reviewer #1: Yes

Reviewer #2: Yes

4. Have the authors described where all data underlying the findings will be made available when the study is complete?

Reviewer #1: No

Reviewer #2: Yes

5. Is the manuscript presented in an intelligible fashion and written in standard English?

Reviewer #1: Yes

Reviewer #2: Yes

6. Review Comments to the Author

You may also provide optional suggestions and comments to authors that they might find helpful in planning their study.

Reviewer #1: The authors propose a follow up study of anchoring effects on research evaluation. In their previous study they found an anchoring effect of citation impact. This study will test the latter treatment on the same set of papers (instead of separate sets like the old study did) and improve the feasibility of a causal explanation. This investigation will inform us on the efficacy of research evaluation. Furthermore, they aim to test the effect of irrelevant anchors in this context more precisely with an extra questionnaire item following more closely the methodology of Tversky and Kahneman’s classic study. This will possibly be yet another test of this widely shown effect.

The proposal describes sufficiently the experimental design, the questionnaire and instruments for the test, and the statistical framework. The authors also perform a power analysis to justify their sampling approach.

Although the goals and means are clear, it is important to reflect on the representativity of the study. The study problematizes science evaluation, but it only represents a special group of scientific papers, namely the highest impact. The response rate of the previous study is low (“About 2% reached the last page of their survey.”), which makes the authors sample only the top 1% of citation impact within the given subject category, in order to ensure that they will receive enough responses. The technical difficulty of evaluating low impact papers (0 impact papers are impossible to sample with the experimental design) puts a limitation on the study, and it seems to be the main reason the study is restricted. This limitation should be acknowledged in the paper, and perhaps the experiment should try to overcome this restriction somehow expanding the sample space beyond the top 1%.

Reviewer #2: I have reviewed the manuscript entitled "Are Quality Assessments in Science

Affected by Anchoring Effects? A Proposal for a Follow-up Study," authored by

Bornmann and Ganser and submitted for publication in PLOS ONE.

This manuscript represents a "Registered Report Protocol" in which the authors

present a comprehensive plan to investigate the presence of anchoring effects

in the assessment of research papers. The rationale behind this study is

well-grounded, and the anticipated outcomes appear highly achievable,

especially in light of a recent publication by these authors in PLOS ONE [Ref.

2]. The authors intend to administer a questionnaire to corresponding authors

of articles chosen from the Web of Science (WOS). These participants will be

assigned to various experimental conditions meticulously outlined in the

manuscript to ascertain whether citation counts exert an anchoring influence

on quality evaluations. These diverse conditions will serve as control groups,

as meticulously elucidated in the manuscript. Furthermore, all data will be

anonymized to ensure the confidentiality of respondents.

I find the research design to be ingeniously conceived, and it is likely to

yield valuable insights. The manuscript is well-written, and the research

problem is persuasively presented. Given that scientific progress hinges

significantly on the evaluation of research processes and that the discovery

of substantial anchoring effects associated with bibliometrics could imply

that these metrics shape the societal order they are designed to gauge, I

warmly endorse the publication of this manuscript in its current form.

7. PLOS authors have the option to publish the peer review history of their article (what does this mean?). If published, this will include your full peer review and any attached files.

Reviewer #1: No

Reviewer #2: No

---

## [Author Response · Author response to Decision Letter 0]

11 Oct 2023

We would like to express our thanks to the reviewers for their positive statements on our manuscript and the recommendations for improving the paper. Besides targeting the reviewer’s comments, we downloaded a new dataset from our in-house databases for this study since we received a new dataset release recently. This results in the change of some numbers in the preregistrations.

Reviewer 1:

P1) The authors propose a follow up study of anchoring effects on research evaluation. In their previous study they found an anchoring effect of citation impact. This study will test the latter treatment on the same set of papers (instead of separate sets like the old study did) and improve the feasibility of a causal explanation. This investigation will inform us on the efficacy of research evaluation. Furthermore, they aim to test the effect of irrelevant anchors in this context more precisely with an extra questionnaire item following more closely the methodology of Tversky and Kahneman’s classic study. This will possibly be yet another test of this widely shown effect. The proposal describes sufficiently the experimental design, the questionnaire and instruments for the test, and the statistical framework. The authors also perform a power analysis to justify their sampling approach.

Ad P1) Thank you! Note that we reserved a DOI where the reader of the upcoming paper will find the anonymized data.

P2) Although the goals and means are clear, it is important to reflect on the representativity of the study. The study problematizes science evaluation, but it only represents a special group of scientific papers, namely the highest impact. The response rate of the previous study is low (“About 2% reached the last page of their survey.”), which makes the authors sample only the top 1% of citation impact within the given subject category, in order to ensure that they will receive enough responses. The technical difficulty of evaluating low impact papers (0 impact papers are impossible to sample with the experimental design) puts a limitation on the study, and it seems to be the main reason the study is restricted. This limitation should be acknowledged in the paper, and perhaps the experiment should try to overcome this restriction somehow expanding the sample space beyond the top 1%.

Ad P2) We agree and included a corresponding statement in the manuscript (see page 10). Expanding the sample space slightly would inflate the database underlying the survey without substantially increasing the representativity. Expanding the sample space further would undermine the goal of receiving enough responses for each paper, because the distribution of citations is very skewed. Therefore, we prefer to stick to the 1%-sample.

Reviewer 2:

Thank you for the positive feedback!

---

## [Editor Report · Decision Letter 1]

16 Oct 2023

Are quality assessments in science affected by anchoring effects? The proposal of a follow-up study

PONE-D-23-23451R1

Dear Dr. Bornmann,

We’re pleased to inform you that your manuscript has been judged scientifically suitable for publication and will be formally accepted for publication once it meets all outstanding technical requirements.

Kind regards,

Filipi Nascimento Silva

Academic Editor

PLOS ONE

Additional Editor Comments (optional):

The authors addressed all the points raised by the reviewers by providing a plan for data sharing and mentioning the limitations of the study in case of a low survey response rate.